# Antibacterial activity of essential oils from Ethiopian *thyme* (*Thymus serrulatus* and *Thymus schimperi*) against tooth decay bacteria

**Destaw Damtie** [1] *, **Yalemtsehay Mekonnen** [2]

**1** Department of Biology, Bahir Dar University, Bahir Dar, Amhara Regional State, Ethiopia, **2** Department of Microbial Cellular and Molecular Biology, Addis Ababa University, Addis Ababa, Ethiopia

* zegades96@gmail.com

**Data Availability Statement:** All data generated or analyzed during this study except the chemical compositions of T. serrulatus and T. schimperi EOs are included within the manuscript and its

## Abstract

In this study, we evaluated the antibacterial activities of the essential oils (EOs) of *Thymus serrulatus* and *Thymus schimperi* collected from Ofla (Ofl), Alamata (Ala), Yilmana Densa (Yil), Tarmaer (Tar), Butajira (Buta), and Bale (Bal) in Ethiopia against cariogenic bacteria (*Streptococcus mutans* and *Lactobacillus*) isolated from human teeth. Inhibition zones (IZs), minimum inhibitory concentrations (MICs), and minimum bactericidal concentrations (MBCs) were measures of the antibacterial activity. Significant bacterial inhibitions resulted in a dose-and EO-dependent manner. At 128 μl/mL, IZs against *S. mutans* were 37.33 mm (Tar), 36.00 mm (Bal), 33.67 mm (Yil), 33.33 mm (Ofl), 30.00 (Ala), and 29.67 mm (Buta) and IZs against *Lactobacillus* were 31.00 mm (Tar), 30.67 mm (Yil), 27.67 (Bal), 27.00 (Buta), 26.67 (Ofl), and 21.33 (Ala). The respective inhibition zones due to 3% DMSO (negative control) and 3% $H_2O_2$ (positive control) were 0.00 mm/30.00 mm against *S. mutans* and 0.00 mm/29.00 mm against *Lactobacillus*. At 128 μl/mL dose, all the EOs resulted in significantly higher inhibition zones than that of 3% $H_2O_2$ against *S. mutans* and *Lactobacillus*.

## Introduction

Dental caries is one of the most prevalent infectious diseases of man [1]. It is a biofilm-mediated, sugar-driven, multifactorial, dynamic disease that results in the phasic demineralization and remineralization of dental hard tissues [2]. Worldwide, approximately 2.43 billion people (36% of the population) had dental caries in their permanent teeth [3]. Reports also show that there is a marked global increase in the prevalence of dental caries [4].

Dental caries is a chronic endogenic multifactorial bacterial infection with gram-positive mutans *Streptococci* and *Lactobacilli* long being recognized as the primary cariogenic organisms [5]. Key organisms which cause dental caries include *Streptococcus* species (*S. sanguinis*, *S. mitis* and *S. crista*), *Lactobacillus* species (*L. gasseri*, *L. fermentum*, and *L. salivarius*), *Fusobacterium*, *Bacteroides*, *Porphyromonas*, *Prevotella*, *Neisseria*, *Veillonella*, *Corynebacterium*, *Actinomyces*, and *Treponema* species [6].

supplementary information files. The chemical compositions of the EOs of these species are published in the Journal of Essential Oil Research as noted in the paper and References.

**Funding:** The authors received no specific funding for this work.

**Competing interests:** The authors declare that they have no competing interests.

**Abbreviations:** Ala, Essential Oil of thyme from Alamata; ANOVA, Analysis of Variance; Bal, Essential Oil of thyme from Bale; Buta, Essential Oil of thyme from Buta Jira; DMFT, Decayed, Missing and Filled Teeth; DMSO, dimethyl sulfoxide; EOs, Essential Oils; EUCAST, European Committee for Antimicrobial Susceptibility Testing; MBC, Minimum Bactericidal Concentration; MIC, Minimum Inhibitory Concentration; MRS agar, de Man, Rogosa and Sharpe agar; MS agar, Mitis Salivarius agar; Ofl, Essential Oil of thyme from Ofla; SPSS, Statistical Package for Social Sciences; Tar, Essential Oil of thyme from Tarmaber; Yil, Essential Oil of thyme from Yilmana Densa.

Oral hygiene, pit and fissure sealing, substituting xylitol for sucrose, vaccines for MS bacteria, oral health management of the primary caregiver of children, and fluoride application are the main methods of caries prevention [7]. However, fluoride applications may lead to elevated levels of fluoride in the body [8].

Owing to cultural and natural reasons, the demand for herbal toothpaste is increasing from time to time. Consequently, the sale of herbal products is exceeding fluoride-based toothpaste [9]. Besides, herbal kinds of toothpaste are reported to be more efficient in the maintenance of oral hygiene and gum bleeding as compared to the non-herbal toothpaste [10].

Furthermore, the growing resistance to classic antimicrobial drugs is a global problem [11]. These are driving forces for the efforts done on the screening of natural products like plant EOs to discover new antimicrobial agents [12]. Many *in vitro* studies show that the EOs of *Achillea ligustica*, *Baccharis dracunculifolia*, *Croton cajucara*, *Cryptomeria japonica*, *Coriandrum sativum*, *Eugenia caryophyllata*, *Lippia sidoides*, *Ocimum americanum*, and *Rosmarinus officinalis* had promising effects on caries-related streptococci (mainly *S. mutans*) and lactobacilli [13].

*Lamiaceae* is one of the most diverse and widespread plant families with antimicrobial properties [14]. In Ethiopia, this family is represented by over 20 genera *Thymus* and *Ocimum* being the two most known [15]. The genus *Thymus* is represented by (*T. serrulatus* and *T. schimperi*) both of which are indigenous to Ethiopia [16]. Studies in these species revealed that they have antibacterial activities. The chloroform extract of *T. schimperi* resulted in inhibition of clinical isolates of human pathogenic bacteria including the Methicillin-Resistant *S. aureus* (MRSA) [17]. The essential oil of the same species also had marked antibacterial activities [18]. *Thyme* EOs are known antibacterials [19] against both Gram-positive and Gram-negative bacteria. For example, EO of *T. vulgaris* was found effective against *S. aureus*, *S. enteritidis*, *E. coli*, and *P. aeruginosa* [14] and extracts of *T.vulgaris* against *H. pylori* and *S. mutans* [20, 21]. The antibacterial activity of *thyme* EOs is associated with their different constituents [22]. The effectiveness of the constituents is based on the functional groups they possess and can be ranked as phenols, aldehydes, ketones, alcohols, esters, and hydrocarbons in descending order [23].

The Ethiopian *thyme* species are known to possess carvacrol and thymol as their major components [16, 24, 25] and are claimed to have antibacterial and antifungal properties [26]. The major chemical constituents of *T. serrulatus* and *T. schimperi* in the present study are carvacrol followed by thymol [25]. Thus, the present study was aimed to test the *in vitro* antibacterial activities of *T. serrulatus* and *T. schimperi* EOs against the isolated *Streptococci* and *Lactobacilli*.

## Materials and methods

### Plant material collection and identification

Aerial parts of *T. serrulatus* and *T. schimperi* were collected between July and September 2013 Alamata (37p- 0542153N, 1369589E) and Ofla (37p- 0558356N, 1389146E) (Tigray Region), Yilmana Densa (37p- 0325366N, 1228943E) and Tarmaber (37p- 0580250N, 1089386E) (Amhara Region), Butajira (37p -0420973N, 897876E) (Southern Nations Nationalities and Peoples Region), and Dinsho (37p- 0586113N, 785143E) (Bale, Oromia Region). These plants are wild plants which were found in communal lands of the study sites. Besides, these plants are not endangered protected species. Therefore, specific permission was not required for the collection and investigation of the mentioned plants.The Thymus species were identified by Mr. Melaku Wondafrash, a botanist working in the National Herbarium of Addis Ababa University, Addis Ababa, Ethiopia, and voucher specimens were deposited in the Herbarium.

## Preparation of plant materials and extraction of essential oils

The collected *T. serrulatus* and *T. schimperi* plants were shade dried and milled into powders. The fine powders (200 g) of each plant were added to 2 L distilled water (with vegetal material/ extraction solvent rate = 1/10 (w/v) in a 4 L round bottom glass flask and subjected to water distillation for 3 h using Clevenger type apparatus. Then the volume of each oil was quantified in milliliters (mL), dried over anhydrous sodium sulfate and stored in dark glass at 4˚C until used.

## Collection of unstimulated whole saliva

**Inclusion and exclusion criteria.**    Students from the biology department of Addis Ababa University were given a full explanation about the purpose of the research. Voluntary students signed letters of consent which were approved by the Research Ethics Review Committee of the College of Natural Sciences of Addis Ababa University. Those students with a DMFT (Decayed, Missing, and Filled Teeth) score of zero were included as subjects to donate saliva. On the other hand, students who were not voluntary to give saliva, those who took antibiotics or corticoids in the last 20 days, who have the habit of smoking, with mucosal inflammatory lesions, and who took orthodontic treatment were excluded. The following method adapted from Navazesh & Kumar [27] was used to collect saliva. The subjects were advised to refrain from intake of any food or beverage one hour before the test session. Smoking, chewing gum, and intake of coffee was also prohibited during this hour. They were advised to rinse their mouth several times with distilled water and then to relax for five minutes. They were then told to swallow to void the mouth of saliva. Then they were told to lean their heads forward over the test tube. They were told to slightly open their mouths and allow saliva to drain into the tubes. At the end of the collection period, they were asked to collect any remaining saliva in their mouths and spit it into the test tubes. Besides, the subjects were asked to brush their teeth using sterile swabs and put them into the test tubes containing their saliva.

## Isolation and identification of cariogenic bacteria

**Culture method.**    Saliva was vortex mixed and ten-fold diluted with sterile water before plating. Mitis Salivarius agar (MS agar) supplemented with potassium tellurite and de Man, Rogosa, and Sharpe agar (MRS agar) were used as culture media for isolation of total *Streptococci* and *Lactobacilli* respectively. Appropriate amounts of semi-solid media were powered on to Petri-plates and were allowed to solidify at room temperature. To each plate, 100 μL of the diluted sample was spread evenly. The plates were collected and sealed in an anaerobic jar. The anaerobic atmosphere in the anaerobic jar was created using candlelight. Then the plates in the anaerobic jar were incubated at 37˚c for 48 hours.

**Identification and confirmation of salivary *Streptococci* and *Lactobacilli*.**    To isolate salivary streptococci especially *S. mutans*, two bacitracin disks that contain 5μl bacitracin were placed 2cm apart on the inoculated agar. Bacterial colonies were observed under a dissecting microscope and deduced based on colony morphology, shape, and color. Furthermore, a confirmation for *S. mutans* and *Lactobacilli* was done by Gram staining, catalase tests, arginine dehydrogenase test, and sugar fermentation tests (mannitol, sorbitol, inulin, lactose, raffinose, and melibiose). Aerobic growth was also tested by inserting inverted Durham tubes into test tubes containing broths where the bacteria were inoculated. The presence or absence of air bubbles at the bottom of inverted Durham tubes was considered as an indicator of aerobic or anaerobic respiration respectively [27] "S1–S3 Figs".

**Biochemical tests to determine salivary *Streptococci* and *Lactobacilli*.**    These bacteria were allowed to grow in Tryptone Soy Broth containing 0.1% carbohydrates (raffinose, inulin,

mannitol, melibiose, lactose, and sorbitol) and 0.5% arginine. Besides, 7.2mL/L of 0.25% phenol red was used as an indicator for the fermentation of the carbohydrates and arginine by the bacteria. The formation of yellow and reddish-pink colors respectively after incubation for 48hrs was considered as positive and negative results of carbohydrate fermentation [28]. On the other hand, the negative and positive fermentation results towards arginine were indicated by the formation of yellow or red and purple colors respectively at the end of 48hrs incubation "S4 Fig".

## Determination of the antibacterial activities of the EOs

Susceptibility tests against *S. mutans* and *Lactobacilli* were accomplished using the disk diffusion method on MS agar [29, 30] and MRS agar respectively. The essential oils were diluted in 3% DMSO (DMSO/Distilled water; vol/vol) to the doses of 16 μL/mL, 32 μL/mL, 64 μL/mL, and 128 μL/mL. Paper disks (6 mm diameter) immersed in each dose were allowed to dry at room temperature and placed on Petri-plates inoculated with salivary *S. mutans* and *Lactobacilli*. Other disks immersed in 3% DMSO and 3% $H_2O_2$ were also used as negative and positive controls respectively.

**Determination of minimum inhibitory concentration (MIC) and minimum bactericidal concentration (MBC).** The MIC was determined using the agar dilution method which is described by the European Committee for Antimicrobial Susceptibility Testing (EUCAST) of the European Society of Clinical Microbiology and Infectious Diseases [31]. To determine the MIC, 20mL volumes of agar were used in 9cm Petri dishes for agar dilution. Nineteen mL volumes of each molten agar (MS and MRS agars) were added separately to 1mL volumes of each EO dose to make the total volume 20mL. The overall procedure followed is explained in the next paragraph.

MS and MRS agars prepared as recommended by the manufacturers were set to cool to 50˚C in a water-bath. EOs of *thyme* were prepared into doses of 0.25 μL/mL, 0.5 μL/mL, 1 μL/mL, 2 μL/mL, and 4 μL/mL in 25-30mL containers (five containers for each selective medium). Nineteen mL of molten agars were added to each container and mixed thoroughly, and finally poured into pre-labeled sterile Petri dishes on a level surface. The plates were allowed to dry at room temperature to avoid drops of moisture on the surface of the agar. Bacterial suspensions grown in Tryptic Soy broths supplemented with 0.2% glucose [32, 33] and incubated for 48hrs were inoculated on the dry plates. The inoculum spots were then allowed to dry at room temperature before inverting the plates for incubation. Finally, the plates were incubated at 37˚C in anaerobic jars for 48 h. The MIC (the lowest concentration of the extracts that completely inhibited visible growth) was judged by the naked eyes. MBC was determined by taking scratches from MIC tests and trying to grow bacteria on new agar plates. In the case of the growth of colonies from scratches taken from the MIC test plates, the MIC was not considered as MBC and vice-versa.

## Statistical analysis

The data were expressed as the mean ± SEM for each group. A computer program (SPSS version 20) was used for statistical analysis. The differences among the groups were performed using one-way analysis of variance (ANOVA) followed by LSD postHoc multiple comparisons. P values < 0.05 were considered statistically significant.

## Ethical consideration

This study was conducted after the necessary ethical clearance was obtained from the Ethical Committee of College of Natural Sciences, Addis Ababa University "S1 File".

**Table 1. Biochemical and physical test results against *S. mutans* and *Lactobacillus* collected from saliva.**

| Bacterium | Sugar/Amino Acid fermentation | | | | | | | Cat. test | Gas from Glu | Gram stain | Cell shape | Bacitracin test |
|---|---|---|---|---|---|---|---|---|---|---|---|---|
| | Raf | Man | Mel | Lac | Arg | Inu | Sor | | | | | |
| *Streptococci* | + | + | + | + | - | + | + | - | - | + | Spherical | - |
| *Lactobacilli* | + | + | + | + | - | + | + | - | - | + | Cylindrical | + |

Raf = Raffinose, Man = Mannitol, Mel = Melibiose, Arg = Arginine, Inu = Inulin, Cat = Catalase, Glu = Glucose, (+) = able to ferment or breakdown, gram-positive, bacitracin sensitive, (-) = not able to ferment or break down or do not produce gas from glucose fermentation, gram-negative, resistant to bacitracin.

## Results

### Isolation of *S. mutans* and *Lactobacilli*

*S. mutants* and *Lactobacillus* grew as separate colonies on selective media (MS and MRS) agars respectively. Both isolates produced yellow colors by fermenting the sugars raffinose, mannitol, melibiose, lactose, inulin, and sorbitol. Both isolates were grown in broths supplemented by arginine produced brick red color. They too were catalase-negative and they were unable to generate gas from glucose. The gram stain of the isolates revealed gram-positive spherical chains (*Streptococci*) and cylindrical cells (*Lactobacilli*). Furthermore, *Streptococcus* was found to be resistant to Bacitracin discs (5μl/disc) "S5–S18 Figs". The overall biochemical and physical test results of salivary isolates are presented in Table 1.

### Antibacterial tests of *thyme* EOs against *S. mutans* and *Lactobacilli*

**Antibacterial activity of the EOs against *Lactobacillus*.** The EOs resulted in a dose-dependent inhibition of *Lactobacillus* of which Tar was the best followed by Yil, Bal, Ofl, Buta, and Ala in decreasing order (Fig 1). The 16 μl/mL dose of these EOs against *Lactobacillus*

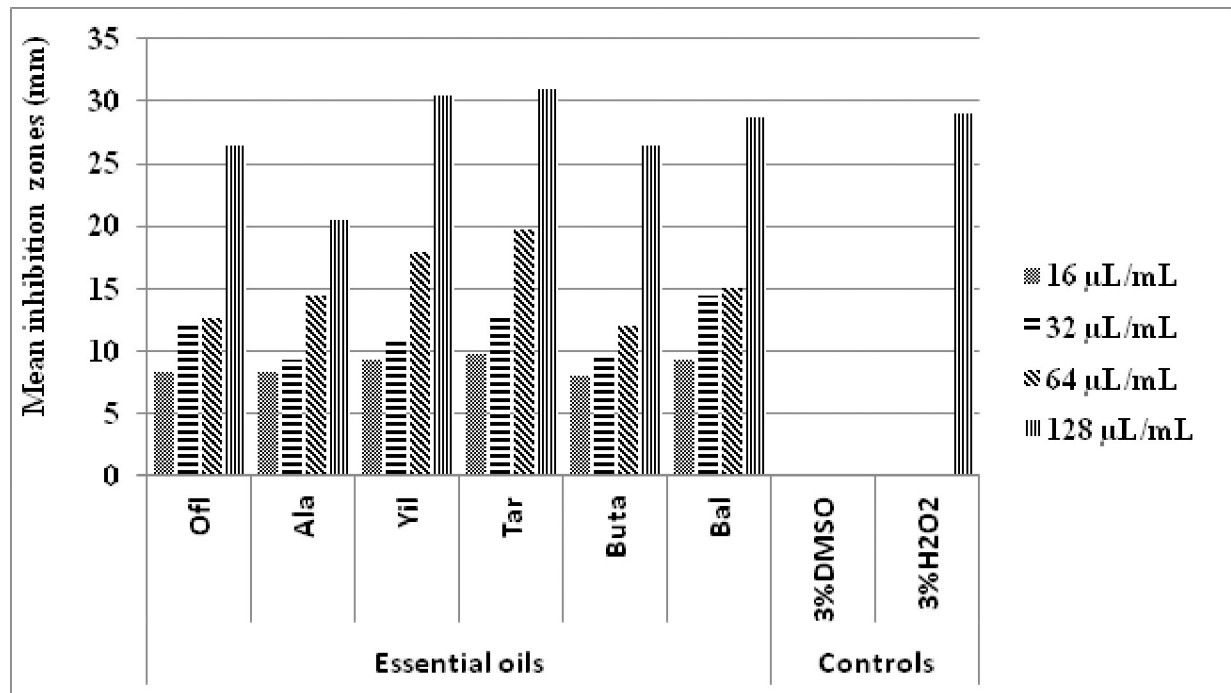

**Fig 1. Inhibition of *Lactobacillus* by EOs from six localities with doses ranging from 16 μl/mL to 128 μl/mL in comparison with the negative and positive controls.**

resulted in inhibition zones close to 10 mm. Correspondingly, the 32 µl/mL doses except that of Buta and Ala resulted in inhibition zones higher than 10 mm. The trend of increase is also visible at 64 µl/mL doses which resulted in inhibition zones between 10 mm and 20 mm in which Buta was the least close to 10 mm and Tar was the highest close to 20 mm. At the dose of 128 µl/mL, Tar and Yil resulted in inhibition zones higher than that of the positive control (3% $H_2O_2$) while Bal and Ofl resulted in inhibition zones closer to this control. However, the inhibition zone recorded by 128 µl/mL EO of Ala was lower than that of the positive control. In the same way, the EOs doses ranging from 16 µl/mL to 128 µl/mL of Ala showed considerable inhibition to this bacterium even though their zones of inhibition were less than that of the 3% $H_2O_2$. The negative control (3% DMSO) on the other hand did not inhibit *Lactobacillus*.

The differences in mean inhibition zones of *Lactobacillus* are indicated in Table 2. The highest inhibitions of *Lactobacillus* that resulted from the 128 µl/mL doses of Yil and Tar were significantly higher than similar doses of Ofl, Ala, Buta, Bal EOs, and the positive control (3%

**Table 2. Antibacterial activity of the EOs of Ofl, Ala, Yil, Tar, Buta, and Bal using agar disc diffusion method compared to the negative and positive controls (mean ± SEM for three repetitions)\*.**

| Treatments | Concentration | Bacteria | |
|---|---|---|---|
| | | *S. mutans* | *Lactobacillus* |
| | | Mean ± SEM | Mean ± SEM |
| Controls | 3% DMSO | 0.00 ± 0.00 [o] | 0.00 ± 0.00 [m] |
| | 3% $H_2O_2$ | 30.00 ± 1.15[d] | 29.00 ± 1.00[b] |
| Ofl | 16 µl/mL | 9.67 ± 0.33[m] | 8.33 ± 0.67[kl] |
| | 32 µl/mL | 11.67 ± 0.33[l] | 10.33 ± 0.33[ij] |
| | 64 µl/mL | 17.00 ± 0.58[ij] | 12.67 ± 0.33[h] |
| | 128 µl/mL | 33.33 ± 0.33[c] | 26.67 ± 0.33[c] |
| Ala | 16 µl/mL | 9.00 ± 0.58[mn] | 8.33 ± 0.33[kl] |
| | 32 µl/mL | 12.67 ± 0.67[l] | 9.33 ± 0.33[jk] |
| | 64 µl/mL | 22.33 ± 0.33 [f] | 14.33 ± 0.33[g] |
| | 128 µl/mL | 30.00 ± 0.58[d] | 21.33 ± 0.33 [d] |
| Yil | 16 µl/mL | 8.00 ± 0.00[n] | 9.33 ± 0.33[jk] |
| | 32 µl/mL | 14.33 ± 0.33[k] | 10.67 ± 0.33[i] |
| | 64 µl/mL | 25.00 ± 0.00[e] | 18.00 ± 0.00[f] |
| | 128 µl/mL | 33.67 ± 0.33[c] | 30.67 ± 0.33[a] |
| Tar | 16 µl/mL | 14.33 ± 0.89[k] | 9.67 ± 0.33[ij] |
| | 32 µl/mL | 18.67 ± 0.67[h] | 10.33 ± 0.33[ij] |
| | 64 µl/mL | 26.00 ± 1.00 [e] | 19.67 ± 0.88[e] |
| | 128 µl/mL | 37.33 ± 0.33[a] | 31.00 ± 0.58[a] |
| Buta | 16 µl/mL | 9.67 ± 0.33[m] | 8.00 ± 0.00[l] |
| | 32 µl/mL | 17.67 ± 0.33[hi] | 9.67 ± 0.33[ij] |
| | 64 µl/mL | 20.33 ± 0.33[g] | 12.00 ± 0.58 [h] |
| | 128 µl/mL | 29.67 ± 0.33[d] | 27.00 ± 0.58[c] |
| Bal | 16 µl/mL | 9.67 ± 0.00[m] | 9.33 ± 0.33[jk] |
| | 32 µl/mL | 16.00 ± 0.58[j] | 12.33 ± 0.33[h] |
| | 64 µl/mL | 20.00 ± 1.00[g] | 15.00 ± 0.00[g] |
| | 128 µl/mL | 36.00 ± 0.58[b] | 27.67 ± 0.33[c] |

Notes: Ofl = Ofla, Ala = Alamata, Yil = Yilmana Densa, Tar = Tarmaer, Buta = Butajira, Bal = Bale, 3% DMSO = negative control-did not show any activity; 3% $H_2O_2$ = Positive control, Least significant difference for *S. mutans* = 1.1861, Least significant difference for Lactobacillus = 1.0757, Means with the same letters in the same column are not significantly different (p>0.05)

\* = the strength of activity is presented as resistant (> 7 mm), intermediate (>12 mm) and susceptible (> 18 mm) [34].

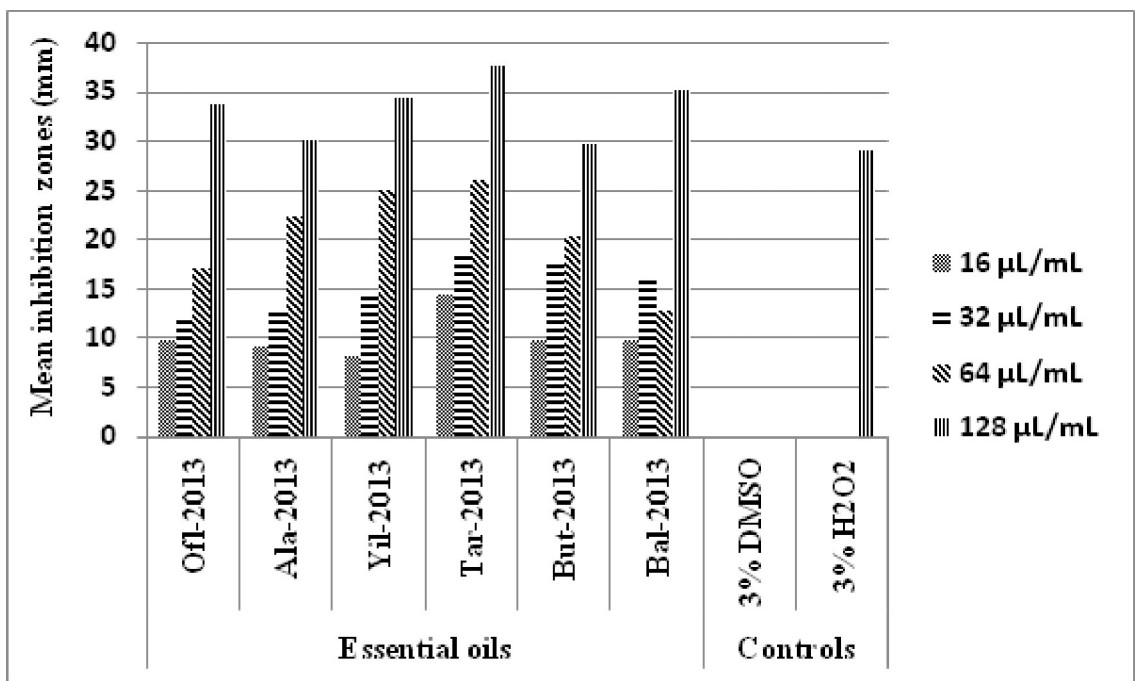

**Fig 2. Inhibition of *S. mutans* by EOs from six localities with doses ranging from 16 μl/mL to 128 μl/mL in comparison with the negative and positive controls.**

$H_2O_2$). On the other hand, 3%$H_2O_2$ inhibited this bacterium with inhibition zones significantly higher than the 16, 32, and 64 μl/mL doses of all EOs and the 128 μl/mL doses of Ofl, Ala, Buta, and Bal EOs. The inhibition zone of *Lactobacillus* by the negative control (3% DMSO) was significantly lower than that of all the doses of the EOs and the positive control. Thus the highest and lowest significant mean inhibition zones against *Lactobacillus* resulted from 128 μl/mL EO of Tar and 16 μl/mL of Buta respectively. Generally, the different doses of Tar and Yil EOs were found to be the strongest EOs against *Lactobacillus*.

**Antibacterial activity of the EOs against *S. mutans*.** As can be depicted from Fig 2, *S. mutans* were inhibited by the different doses of the EOs; Ofl, Ala, Yil, Tar, Buta, and Bal in a dose-dependent manner. This bacterium was inhibited by 3% $H_2O_2$ (positive control) and was not inhibited by 3% DMSO (negative control). The smallest range of inhibition of this bacterium was due to the 16 μl/mL concentration of all the EOs which was between 8 and 10 mm. The mean minimum inhibition zones of all the test EOs at the dose of 32 μl/mL lied between 10 and 20 mm Ofl being with the least (11.67 ± 0.33 mm) and Tar with the highest (18.67 ± 0.67) inhibition zones.

At 64 μl/mL concentration, *S. mutans* was inhibited with mean inhibition zones better than that resulted due to the 32 μl/mL concentration. At this dose, four of the EOs Ala, Yil, Tar, and Buta resulted in mean inhibition zones between 20 and 30 mm and Tar resulted in the highest mean inhibition zone (26.00 ± 1.00) and Buta the lowest (20.33 ± 0.33). The rest two EOs Ala and Bal, however, resulted in mean inhibition zones lower than 20 mm. At 128 μl/mL concentration of the EOs, *S. mutans* was highly inhibited by all the EOs. At this dose level, around 67% of the EOs (Ofl, Yil, Tar, and Bal) inhibited *S. mutans* with inhibition zones higher than that of the positive control. The rest 33% of the EOs (Ala and Buta) inhibited this bacterium with inhibition zones comparable to that of the positive control.

At the dose 128 μl/mL, Tar inhibited *S. mutans* with inhibition zones significantly higher than the rest EOs and the positive control. It was followed by the 128 μl/mL dose of Bal which

in turn was followed by the 128 µl/mL doses of Ofl and Yil. The least inhibition of *S. mutans* at this dose was seen in Ala and Buta EOs. At 64 µl/mL dose, significant inhibition of *S. mutans* was as follows: Yil and Tar > Ala >Buta and Bal. At the dose of 32 µl/mL, the order of inhibition of *S. mutans* was Tar and Buta > (significantly higher than) Ala > Bal > Yil > Ofl. The 16 µl/mL dose of Tar inhibited *S. mutans* with inhibition zone comparable with the 32 µl/mL of Yil and with significantly higher inhibition zone than the 32 µl/mL of Ala > 16 µl/mL of Ofl, Ala, Buta and Bal > 16 µl/mL of Yil > 3%DMSO.

Thus the Tar (128 µl/mL) and the Yil (16 µl/mL) were found to be the most and the least inhibitor doses of *S. mutans*. The positive control resulted in inhibition zones higher than the 16, 32, and 64 µl/mL doses of the entire test EOs and the negative control (3% DMSO) resulted in no inhibitions. In general, Tar EO appeared to be the strongest inhibitor of *S. mutans* than the rest EOs Table 2.

**MIC and MBC of EOs against *S. mutans* and *Lactobacillus*.** The MIC and MBC of the test EOs were determined for both bacteria. The MIC and MBC of *S. mutans* were found to be 0.25 µl/mL by Bal and 0.5 µl/mL by all the rest EOs Table 3. On the other hand, the MIC and MBC against *Lactobacillus* were at the dose of 0.5 µl/mL by all the test EOs.

## Discussion

According to different findings, oxygenated monoterpenes like thymol and carvacrol exhibit strong antimicrobial activity and hydrocarbon derivatives possess lower antimicrobial properties. This is because the low water solubility of hydrocarbons limits their diffusion through the medium. For example, in the work by Sokovic et al. [35], the trend of antibacterial activity of EO components were as; linalyl acetate < limonene < β-pinene < α-pinene < camphor < linalool < 1,8-cineole < menthol < thymol < carvacrol against *Proteus mirabilis* (human isolate) *Pseudomonas aeruginosa* (ATCC 27853), *Staphylococcus aureus* (ATCC 25923), *Staphylococcus epidermidis* (ATCC 12228), *Micrococcus flavus* (ATCC 9341), *Bacillus subtilis* (ATCC 10707), *Escherichia coli* (ATCC 0157:H7), *Enterobacter cloacae* (human isolate), *Salmonella enteritidis* (ATCC 13076), *Salmonella typhimurium* (ATCC 13311).

The EOs of *thyme* in the present study contained components with high antibacterial activities (thymol and carvacrol) [36]. The respective percentages of thymol and carvacrol in the test EOs were as follows: Ofl (49.55%, 36.34%), Ala (65.63%, 6.68%), Yil (6.52%, 80.84%), Tar (48.84%, 42.12%), Buta (15.77%, 71.83%), and Bal (53.57%, 34.55%) [25]. Therefore, it is possible to infer from this finding that the EOs of Yil, Buta, and Tar showed the highest antibacterial activity against *S. mutans* and *Lactobacillus* followed by that of Bal, Ofl, and Ala respectively. However, minor deviations were seen in the recorded results.

**Table 3. Antibacterial activity (MIC and MBC) of *Thyme* EOs against *S. mutans* and *Lactobacillus*.**

| Botanical Name | Essential Oil | S. mutans | | Lactobacillus | |
| --- | --- | --- | --- | --- | --- |
| | | MIC* (µl/mL) | MBC** (µl/mL) | MIC* (µl/mL) | MBC** (µl/mL) |
| *Thymus serrulatus* | Ofl | 0.5 | 0.5 | 0.5 | 0.5 |
| *Thymus serrulatus* | Ala | 0.5 | 0.5 | 0.5 | 0.5 |
| *Thymus serrulatus* | Yil | 0.5 | 0.5 | 0.5 | 0.5 |
| *Thymus schimperi* | Tar | 0.5 | 0.5 | 0.5 | 0.5 |
| *Thymus schimperi* | Buta | 0.5 | 0.5 | 0.5 | 0.5 |
| *Thymus schimperi* | Bal | 0.25 | 0.25 | 0.5 | 0.5 |

Notes: Ofl = Ofla, Ala = Alamata, Yil = Yilmana Densa, Tar = Tarmaer, Buta = Butajira, Bal = Bale

*MIC = Highest dilution (minimum concentration) showing no detectable growth

**MBC = Highest dilution (minimum concentration) yielded no single colony on a solid medium.

Tar EO showed the highest inhibition of both *S. mutans* and *Lactobacillus*. This may be due to the synergistic/additive activities of thymol and carvacrol components [37] since the two are found in close percentages 48.84% and 42.12%, respectively and the sum of the two is the highest of all the EOs. On the other hand, the least effective was the EO from Ala since it is majorly of thymol (65.63%) and less of carvacrol (6.68%) composition [35].

In the work by Barkat and Bouguerra [38], the inhibition zones of thymol and carvacrol against *S. mutans* were 7.8 mm and 8.0 mm, respectively at 50 mg/mL dose each. This implies carvacrol has more antibacterial activity against *S. mutans* than thymol which agrees with the antibacterial activities of the chemotypes of EOs in the present study. The MIC/MBC values of thymol and carvacrol against *S. mutans* were 5 mg/mL/10 mg/mL and 2.5 mg/mL/5 mg/mL respectively [39]. In the present study, the MIC/ MBC doses were: Ofl, Ala, Yil, and Buta EOs (0.5/0.5 μl/mL) against both *S. mutans* and *Lactobacillus* and Bal EO (0.25/0.25) μl/mL against *S. mutans* and 0.5/0.5 μl/mL against *Lactobacillus*.

The present study shows that the EOs of *T. serrulatus* from Ala, Ofl, and Yil and *T. schimperi* from Tar, Buta, and Bal showed antimicrobial activity against cariogenic bacteria namely *S. mutans* and *Lactobacillus*. *S. mutans* and *Lactobacillus* have been tested for antimicrobial susceptibility to other EOs [39]. As far as known, this is the first report on the antibacterial activities of the EOs of *T. serrulatus* and *T. schimperi* against oral pathogenic bacteria.

*Thyme* EOs have been known to possess antibacterial properties for long [19], and the Gram-positive bacteria are generally susceptible to these EOs than the Gram-negative ones [14]. *S. mutans* and *Lactobacillus* both of which are Gram-positive bacteria thus were highly inhibited by the EOs of *T. serrulatus* and *T. schimperi*. The major components of *thyme* EOs result in antibacterial activities against cariogenic pathogens [39]. *T.serrulates* and *T. schimperi* found to be thymol (Ofl, Ala, Baland Tar) and carvacrol (Yil and Buta) chemotypes [25] in this study thus inhibited *S. mutans* and *Lactobacillus*. This agrees with other *thyme* EOs which have antiseptic effects when applied externally or taken internally [19]. This may be due to the lipophilic character of the hydrocarbon skeletons of the EO constituents as well as the hydrophilic character of their functional groups. The rank of activity of EO components is: phenols > aldehydes > ketones > alcohols > esters > hydrocarbons [23]. The findings in this research show that the major components of the EOs were phenols (thymol or carvacrol) [25]. Thus they had effective antibacterial activities against cariogenic bacteria. The antibacterial activity of these EOs may have resulted from the bioactivities of their major components or the interactions of all their components [40]. For example in a review of studies made by Bassolé & Juliani [41], thymol and carvacrol interactions majorly showed synergistic activities against pathogenic bacteria. Synergistic activities were also seen between combinations of Eos; thymol and eugenol, carvacrol and eugenol, carvacrol and linalool, menthol and geraniol, and menthol and thymol against different bacteria. The total antibacterial activities of EOs, in general, are due to the synergistic, additive, and antagonistic activities of their components.

The mechanism of action of the EOs of *T. serrulatus* and *T. schimperi* against *S. mutans* and *Lactobacillus* may be owing to their action on multiple targets of action [38]. Such sites of action include the ability to inhibit respiration [42, 43], increase the permeability of cytoplasmic and plasma membranes [43, 44], increased potassium ion leakage, increased disruption of the permeability barrier of cell membrane structures, as well as the loss of chemiosmotic control [42, 43]; disruption of polysaccharides, fatty acids, and phospholipids in different layers of these structures [44, 45]. Besides, EOs may cause coagulation of the cytoplasm and damage of lipids and proteins of microorganisms [44]. To sum up, the mode of action of EOs against these bacteria may be in the same way as broad-spectrum membrane-active disinfectants and preservatives, such as phenol derivatives, chlorhexidine, and para benzoic acid derivatives [46]. *S. mutans* and *Lactobacillus* were found to be sensitive to 3% $H_2O_2$.

This is because of their incapacity to produce the enzyme catalase. Catalase converts $H_2O_2$ into harmless molecules, $H_2O$, and molecular $O_2$ [47].

## Conclusion

The EOs of *T. serrulatus* and *T. schimperi* showed a high level of antibacterial activity against tooth decay bacteria, *S. mutans*, and *Lactobacillus* in a dose-dependent manner. The EOs from Tar with close proportions of thymol and carvacrol were the most effective EO followed by the carvacrol dominant EO of Yil and thymol dominant EOs with relatively lower antibacterial activity.

## Supporting information

**S1 Fig.** Isolation of S. mutans (a) and Lactobacillus (b) colonies on MS and MRS agars respectively.
(DOCX)

**S2 Fig.** Results for Gram staining experiments on *S. mutans* (a) and *Lactobacillus* (b).
(DOCX)

**S3 Fig.** Sugar hydrolysis and arginine hydrolysis test results of *S. mutans* (a) and *Lactobacillus* (b).
(DOCX)

**S4 Fig. The aerobic growth test result of *S. mutans* and *Lactobacillus*.**
(DOCX)

**S5 Fig. Inhibition of Lactobacillus by Ofl EO (doses: 16μl/mL, 32μl/mL, 64μl/mL, and 128μl/mL respectively).**
(DOCX)

**S6 Fig. Inhibition of *S. mutans* by Ofl EO (doses: 16μl/mL, 32μl/mL, 64μl/mL, and 128μl/mL respectively).**
(DOCX)

**S7 Fig. Inhibition of *Lactobacillus* by Ala EO (doses: 16μl/mL, 32μl/mL, 64μl/mL, and 128μl/mL respectively).**
(DOCX)

**S8 Fig. Inhibition of *S. mutans* by Ala EO (doses: 16μl/mL, 32μl/mL, 64μl/mL, and 128μl/mL respectively).**
(DOCX)

**S9 Fig. Inhibition of *Lactobacillus* by Yil EO (doses: 16μl/mL, 32μl/mL, 64μl/mL, and 128μl/mL respectively).**
(DOCX)

**S10 Fig. Inhibition of *S. mutans* by Yil EO (doses: 16μl/mL, 32μl/mL, 64μl/mL, and 128μl/mL respectively).**
(DOCX)

**S11 Fig. Inhibition of *Lactobacillus* by Tar EO (doses: 16μl/mL, 32μl/mL, 64μl/mL, and 128μl/mL respectively).**
(DOCX)

**S12 Fig. Inhibition of *S. mutans* by Tar EO (doses: 16μl/mL, 32μl/mL, 64μl/mL, and 128μl/mL respectively).**
(DOCX)

**S13 Fig. Inhibition of *Lactobacillus* by Buta EO (doses: 16μl/mL, 32μl/mL, 64μl/mL, and 128μl/mL respectively).**
(DOCX)

**S14 Fig. Inhibition of *S. mutans* by Buta EO (doses: 16μl/mL, 32μl/mL, 64μl/mL, and 128μl/mL respectively).**
(DOCX)

**S15 Fig. Inhibition of *Lactobacillus* by Bal EO (doses: 16μl/mL, 32μl/mL, 64μl/mL, and 128μl/mL respectively).**
(DOCX)

**S16 Fig. Inhibition of *S. mutans* by Bal EO (doses: 16μl/mL, 32μl/mL, 64μl/mL, and 128μl/mL respectively).**
(DOCX)

**S17 Fig.** Inhibition of *S. mutans* (a) and *Lactobacillus* (b) by 3% DMSO.
(DOCX)

**S18 Fig.** Inhibition of *S. mutans* (a) and *Lactobacillus* (b) by 3% $H_2O_2$.
(DOCX)

**S1 File. Ethical clearance letter.**
(DOCX)

## Acknowledgments

We would like to acknowledge the Ministry of Education of Ethiopia, Bahir Dar University, Addis Ababa University.

## Author Contributions

**Conceptualization:** Destaw Damtie.

**Formal analysis:** Destaw Damtie.

**Investigation:** Destaw Damtie.

**Supervision:** Yalemtsehay Mekonnen.

**Writing – original draft:** Destaw Damtie.

**Writing – review & editing:** Yalemtsehay Mekonnen.

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
