## [Decision Letter · Decision Letter 0]

21 Jul 2020

PONE-D-20-17109

Antibacterial activity of essential oils from Ethiopian thyme (Thymus serrulatus and Thymus schimperi) against tooth decay bacteria

PLOS ONE

Dear Dr. Damtie,

Thank you for submitting your manuscript to PLOS ONE. After careful consideration, we feel that it has merit but does not fully meet PLOS ONE’s publication criteria as it currently stands. Therefore, we invite you to submit a revised version of the manuscript that addresses the points raised during the review process.

We look forward to receiving your revised manuscript.

Kind regards,

Leonidas Matsakas

Academic Editor

PLOS ONE

Journal Requirements:

2. In your Methods section, please provide additional location information of the collection sites, including geographic coordinates for the data set if available.

3. In your Methods section, please provide additional information regarding the permits you obtained for the work. Please ensure you have included the full name of the authority that approved the collection site access and, if no permits were required, a brief statement explaining why.

4. In your Methods section, please provide additional information about the participant recruitment method and the demographic details of your participants. Please ensure you have provided sufficient details to replicate the analyses such as: a) the recruitment date range (month and year), b) a description of any inclusion/exclusion criteria that were applied to participant recruitment, c) a table of relevant demographic details, d) a statement as to whether your sample can be considered representative of a larger population, e) a description of how participants were recruited, and f) descriptions of where participants were recruited and where the research took place.

5.Thank you for stating the following in the Funding Section of your manuscript:

[The Ministry of Education of Ethiopia, Addis Ababa University, and Bahirdar University funded this research work.]

 [No]

Reviewers' comments:

Reviewer's Responses to Questions

**Comments to the Author**

1. Is the manuscript technically sound, and do the data support the conclusions?

Reviewer #1: Yes

Reviewer #2: Yes

2. Has the statistical analysis been performed appropriately and rigorously? 

Reviewer #1: Yes

Reviewer #2: I Don't Know

3. Have the authors made all data underlying the findings in their manuscript fully available?

Reviewer #1: Yes

Reviewer #2: Yes

4. Is the manuscript presented in an intelligible fashion and written in standard English?

Reviewer #1: Yes

Reviewer #2: No

5. Review Comments to the Author

Reviewer #1: The paper “Antibacterial activity of essential oils from Ethiopian thyme (Thymus serrulatus and Thymus schimperi) against tooth decay bacteria” reports the antibacterial activities of the essential oils of Thymus serrulatus and Thymus schimperi collected from different areas of Ethiopia against cariogenic bacteria (Streptococcus mutans and Lactobacillus). The methodology is well described and the results are clearly discussed.

Some points should be clarified before publication:

- The introduction is too short and needs implementing. The other activities of Thymus and EOs in general could be discussed: see for example “Mediterranean essential oils as precious matrix components and active T ingredients of lipid nanoparticles, Int. J. Pharm. 548 (2018) 217–226”.

- Section “Plant material collection and identification”: thymus collected in Ofla is not reported, while it is mentioned in the abstract.

- Line 79: even though a reference was provided, a brief description of the procedure followed to collect saliva should be provided for the benefit of the reader.

Reviewer #2: 1. Essential oil production is not appropriate term. Its better to use professional terms.

2. "Aerial parts of T. serrulatus and T. schimperi collected from the wild ….. ". this sentence needs edition.

3. Please describe the process of dilution of essential oils and pouring them in the agar

4. "They too were not able to release gas from H2O2"

5. "The EOs doses ranging from 16 μl/mL to 128 μl/mL of Ala too showed" needs edition.

The text needs edition grammatically.

6. Its better to define exactly the terms in tables such as : Yil , Tar,Ala ,Buta and ….

7. The same subject is also persistent for the text.

8. "The EOs of thyme in the present study contained components with high antibacterial activities 250(thymol and carvacrol)". With which results, you claimed this in discussion ? please clarify the subject? Or refer to your another published work?

This subject is persistent for all of the discussion.

6. PLOS authors have the option to publish the peer review history of their article (what does this mean?). If published, this will include your full peer review and any attached files.

Reviewer #1: No

Reviewer #2: **Yes: **Shirin Moradkhani

---

## [Author Response · Author response to Decision Letter 0]

28 Aug 2020

Comment 1: Please ensure that your manuscript meets PLOS ONE's style requirements, including those for file naming. The PLOS ONE style templates can be found at

Response 1: Gone through and corrected

Comment 2: In your Methods section, please provide additional location information of the collection sites, including geographic coordinates for the data set if available.

Response 2: Added (Lines 84-87)

Comment 3: In your Methods section, please provide additional information regarding the permits you obtained for the work. Please ensure you have included the full name of the authority that approved the collection site access and, if no permits were required, a brief statement explaining why.

Response 3: Plant materials were collected from the wild where every body used to collect them for local use. Thus, the researchers did not need permit to collect the plant materials. (Lines 90-92)

Comment 4: In your Methods section, please provide additional information about the participant recruitment method and the demographic details of your participants. Please ensure you have provided sufficient details to replicate the analyses such as: a) the recruitment date range (month and year), b) a description of any inclusion/exclusion criteria that were applied to participant recruitment, c) a table of relevant demographic details, d) a statement as to whether your sample can be considered representative of a larger population, e) a description of how participants were recruited, and f) descriptions of where participants were recruited and where the research took place.

Response 4: The participants served only as sources of bacteria. The study was conducted on bacteria. The findings are generalized to bacteria not to sources of bacteria. What was required is Ethical clearance and it is attached with the supplementary materials.

Comment 5: Thank you for stating the following in the Funding Section of your manuscript:

[The Ministry of Education of Ethiopia, Addis Ababa University, and Bahirdar University funded this research work.]

 [No]

Response 5: Edited as [No] (Line 383)

Reviewer 1

Comment 1: 

• The introduction is too short and needs implementing. The other activities of Thymus and EOs in general could be discussed: see for example “Mediterranean essential oils as precious matrix components and active T ingredients of lipid nanoparticles, Int. J. Pharm. 548 (2018) 217–226”.

Response 1: Introduction is lengthened by addition of further information

- Lines 35-62

- Lines 66-59

Comment 2:

• Section “Plant material collection and identification”: thymus collected in Ofla is not reported, while it is mentioned in the abstract.

Response 2:

- Corrected (Line 84)

Comment 3:

• Line 79: even though a reference was provided, a brief description of the procedure followed to collect saliva should be provided for the benefit of the reader.

Response 3:

- Corrected (Lines 110-118)

Reviewer 2

Comment 1

- Essential oil production is not appropriate term. Its better to use professional terms.

Response 1

- Corrected as “Preparation of plant materials and extraction of essential oils” (Line 93)

Comment 2

- "Aerial parts of T. serrulatus and T. schimperi collected from the wild ….. ". This sentence needs edition.

Response 2

- Corrected as “The collected T. serrulatus and T. schimperi plants were shade dried and milled into powders.” (Lines 93-94)

Comment 3

- Please describe the process of dilution of essential oils and pouring them in the agar

Response 3

- Described in Lines 150-152

Comment 4

- "They too were not able to release gas from H2O2"

Response 4

- Modified as “They too were catalase negative” (Line 191)

Comment 5

- "The EOs doses ranging from 16 μl/mL to 128 μl/mL of Ala too showed" needs edition.

Response 5

- Modified as “In the same way, the EOs doses ranging from 16 μl/mL to 128 μl/mL of Ala showed considerable inhibition to this bacterium even though their zones of inhibition were less than that of the 3% H2O2” (Lines 213-215)

Comment 6

- Its better to define exactly the terms in tables such as : Yil , Tar,Ala ,Buta and ….

Response 6

- Corrected (Lines 232 & 278)

Comment 7

- "The EOs of thyme in the present study contained components with high antibacterial activities 250(thymol and carvacrol)". With which results, you claimed this in discussion ? please clarify the subject? Or refer to your another published work?

Response 7

- Corrected (Lines 79, 293, 295, 323, & 329)

---

## [Editor Report · Decision Letter 1]

1 Sep 2020

PONE-D-20-17109R1

Antibacterial activity of essential oils from Ethiopian thyme (Thymus serrulatus and Thymus schimperi) against tooth decay bacteria

PLOS ONE

Dear Dr. Damtie,

Thank you for submitting your manuscript to PLOS ONE. After careful consideration, we feel that it has merit but does not fully meet PLOS ONE’s publication criteria as it currently stands. Therefore, we invite you to submit a revised version of the manuscript that addresses the points raised during the review process.

We look forward to receiving your revised manuscript.

Kind regards,

Leonidas Matsakas

Academic Editor

PLOS ONE

Additional Editor Comments (if provided):

Please address the following:

- Correct the references in the text that were added during the revising process.

- Remove the comments you have included.

- In L145 (the version with corrections indicated), change 'percentage' to '%', add units (i.e. v/v?) and specify specify to which solution was the 3% DMSO prepared.

---

## [Author Response · Author response to Decision Letter 1]

9 Sep 2020

Correct the references in the text that were added during the revising process.

- The newly added references were; 6, 13, 17, 18, 25, 36, and 37. They are found in their correct forms.

Remove the comments you have included.

- Removed.

In L145 (the version with corrections indicated), change 'percentage' to '%', add units (i.e. v/v?) and specify specify to which solution was the 3% DMSO prepared.

- Corrected Line 150

- Line 153

---

## [Editor Report · Decision Letter 2]

14 Sep 2020

Antibacterial activity of essential oils from Ethiopian thyme (Thymus serrulatus and Thymus schimperi) against tooth decay bacteria

PONE-D-20-17109R2

Dear Dr. Damtie,

We’re pleased to inform you that your manuscript has been judged scientifically suitable for publication and will be formally accepted for publication once it meets all outstanding technical requirements.

Kind regards,

Leonidas Matsakas

Academic Editor

PLOS ONE
---

## [Editor Report · Acceptance letter]

29 Sep 2020

PONE-D-20-17109R2 

Antibacterial activity of essential oils from Ethiopian thyme (Thymus serrulatus and Thymus schimperi) against tooth decay bacteria 

Dear Dr. Damtie:

I'm pleased to inform you that your manuscript has been deemed suitable for publication in PLOS ONE. Congratulations! Your manuscript is now with our production department. 

Kind regards, 

on behalf of

Dr. Leonidas Matsakas 

Academic Editor

PLOS ONE